# Enhancing the Intention to Preview Learning Materials and Participate in Class in the Flipped Classroom Context through the Use of Handouts and Incentivisation with Virtual Currency

**Yi-Hsing Chang \*, Jin-Yu Lin and You-Te Lu**

Department of Information Management, Southern Taiwan University of Science and Technology,
Tainan City 71005, Taiwan; ma790105@stust.edu.tw (J.-Y.L.); yowder@stust.edu.tw (Y.-T.L.)
**\*** Correspondence: yhchang@stust.edu.tw

**Abstract:** The flipped classroom approach is an emerging instructional approach that integrates digital technology. This approach has been applied in several fields, and it has demonstrated considerably higher learning effectiveness than conventional teaching modes. Common problems in its implementation that directly or indirectly affect learning effectiveness include students' low intention to preview learning materials and low class participation. To overcome these problems, the present study sought to increase students' intention to preview learning materials and participate in class through the implementation of educational activities integrated with an incentivisation system using a virtual currency and the provision of handouts. Students in two programming classes in the information management department of the participating university were divided into the experimental and control groups. The intention to preview learning materials, levels of class participation, and learning effectiveness were all significantly higher in the experimental group than in the control group.

**Keywords:** flipped classroom; intention to preview learning materials; class participation; incentivisation; virtual currency

## 1. Introduction

Conventional learning practices are characterised by students acquiring the content delivered by teachers. The flipped classroom approach, facilitated by digital technology, reverses the conventional teaching in which the teacher is at the centre of learning and students are passive receivers. Specifically, in the flipped classroom, the teacher guides students to learn actively. For example, teachers prepare lecture videos and online handouts before class that students preview using mobile phones or laptops; this reduces the amount of time spent in class working to understand the material, enabling more time for self-learning, self-reflection on problems encountered, and discussion in class, thereby enhancing students' logical thinking and problem-solving skills.

The flipped classroom approach is an innovative pedagogical model that has begun to be applied in several fields of learning because it can foster self-learning and critical thinking skills and increase learning effectiveness. For example, Akçayır and Akçayır [1] presented a large-scale systematic review of the literature on the flipped classroom to examine its reported advantages and challenges for both students and instructors. A total of 71 research articles were selected for the review from the full range of Social Sciences Citation Indexed journals. The findings reveal that the most frequently reported advantage of the flipped classroom is the improvement of student learning performance. Majon Kumar et al. [2] reported that the grades of low-level students who underwent a flipped classroom intervention were comparable to those of high-level students attending conventional lectures. Guerrero et al. [3] reported improved learning effectiveness in a chemistry class in which a flipped classroom intervention was employed, and found that

students were more interested in learning and solving problems through interaction in class under this educational model. Knežević et al. [4] stated that advanced online courses often focus on actual experience, which is typically neglected in conventional teaching; accordingly, they applied a flipped classroom intervention that involved in-class group discussion, 49.5 h of recorded lecture materials, the provision of briefs, and the administration of online mock examinations and online tests. Simple presentations given before actual in-class implementation helped students successfully accomplish their tasks. Notably, the video materials were used less frequently as the course progressed. This may be attributable to the fact that eight lessons were related to subnetting, a topic which was more difficult for students and required more mental practice. Vasilchenko et al. [5] implemented an innovative educational intervention that combined self-learning and the flipped classroom, allowing for interplay between the two. In the self-learning part, students created their own learning content and learning tools. Students played four main roles, namely those of creator, collaborator, communicator, and learner. Elmaadaway [6] investigated whether the flipped classroom approach improved students' participation and comprehension in Blackboard courses. The results showed that students who learned in a flipped classroom participated more actively in the learning process overall. Wang et al. [7] integrated the flipped classroom approach with problem-based learning through the production of lecture videos and the design of class activities pertaining to problem-based learning. Reminders were provided, and students were encouraged to ask questions. After lecture videos were played, students answered multiple-choice or fill-in-the-blank questions to confirm their understanding of the material. Students who received this flipped classroom intervention had significantly higher grades than those who attended conventional classes.

Regarding the application of the flipped classroom approach to research on programming, Mok and Rao [8] conducted a three-week intensive basic programming course for students attending a preparatory programme at the National University of Singapore. A mixed learning approach was used in the flipped classroom context that included lecture videos, self-assessments, live meetings, in-class lectures, and actual programming activities. Students were divided into groups, and each group was guided by an instructor. Overall, 80.6% of the students passed the course, and more than one-third were deemed capable programmers. Sharp [9] explored whether the application of a flipped classroom approach in a C programming course would increase learning motivation and whether students would recommend flipped learning to their classmates. The results indicated that the flipped classroom approach was more successful than conventional instructional practices. Knutas et al. [10] explored the application of the flipped classroom in university programming courses and constructed a shared structure for course design by using the flipped classroom approach. The flipped classroom approach was more effective than the conventional lecture and exercises method; thus, Knutas suggested that teachers should integrate the flipped classroom into their course designs. Alhazbi [11] explored the application of the flipped classroom approach in programming courses and reported that the flipped classroom approach improved students' learning attitudes and grades. In an investigation of various approaches to course design, Maher et al. [12] found that flipped classroom-based methods were suitable for programming courses, enhancing both learning motivation and learning effectiveness. Moreover, students provided positive feedback on the flipped classroom intervention, which indicated that such educational models could promote learning motivation and active learning.

Researchers have identified students' low intention to preview learning materials and their low levels of class participation as common problems encountered in flipped classroom interventions that directly or indirectly affect learning effectiveness. For example, Majon Kumar et al. [2] noted that some students did not preview the learning materials or make briefs; consequently, their teacher telephoned them to remind them to do so. Guerrero et al. [3] contended that the greatest difficulty in implementing a flipped classroom intervention was that some students did not preview the materials, either because they could not access them, or they forgot to do so. Knežević et al. [4] reported that most

students did not watch the video materials but that the majority of them had sufficiently prepared for class, leading the researchers to speculate that students probably learned the content by using other types of course materials. Alhazbi [11] examined how to encourage students to preview course content.

Mok and Rao [8] indicated that class activities were often conducted in groups, to which certain students could not adapt, causing them to withdraw from the course. Moreover, some students did not participate in class discussions. Vasilchenko et al. [5] reported that some students found the course too difficult and that numerous activities based on group work were not suitable for every student; some students worked more efficiently alone than within a group. In his study of a Blackboard course, Elmaadaway [6] suggested that students are not used to being responsible for their own learning and making more effort, factors which are necessary in the flipped classroom context. He recommended the implementation of strategies such as encouraging students to note down their questions, solving problems during class, and collecting students' opinions and ideas. Wang et al. [7] indicated that insufficient pre-class preparation and low motivation for class participation, as well as low learning efficiency in the classroom, impeded the successful implementation of a flipped classroom approach combined with problem-based learning. Sharp [9] stated that pre-class preparation is necessary for enhancing learning motivation and class participation in a flipped classroom.

Internet banking services enable payment processing and transfer through digital modalities. Electronic money involves the addition of values designated by the government or accounting agencies on antitampering devices. The usage rate of virtual currency, which has emerged with the advancement of encryption technology and competes with diverse forms of currency [13], increases year by year. Examples of virtual currency include Line Coins (used by Line, a type of communication software), points used in online games, and Bitcoin. Virtual currency represents a new form of payment for purchasing commodities and services [14].

STUSTCoin, a type of virtual currency issued by the Southern Taiwan University of Science and Technology in 2019, can be used at 7-Eleven convenience stores in the country as well as at school gymnasiums and cafeterias. STUSTCoin can also be converted into cash vouchers and can be exchanged for selected products at ibon kiosks (at 7-Eleven stores). Students can also freely exchange STUSTCoin among themselves. For each compulsory course, 2000 STUSTCoin units are provided at the beginning of the semester. As the course progresses, the instructor decides the coin distribution method according to student performance, such as their grades or rank in competitions.

The present study explored the effects of the flipped classroom approach and the relevant educational activities and used STUSTCoin units as an incentive to encourage students' intention to preview learning materials, participate in class, and learn effectively. The research questions were as follows:

- Whether incorporating handouts in educational activities and using STUSTCoin as an incentive would significantly increase students' intention to preview learning materials;
- Whether incorporating handouts in educational activities and using STUSTCoin as an incentive would significantly increase their class participation;
- Whether incorporating handouts in educational activities and using STUSTCoin as an incentive would significantly improve their learning effectiveness.

## 2. Methodology

Educational activities were designed according to the main research objectives of increasing students' intention to preview learning materials and their class participation. The study flowchart and the research process, hypotheses, and tools are presented as follows.

*2.1. Research Architecture*

The research architecture, which contains control, independent, and dependant variables, is shown in Figure 1.

(1) Control variables:
- Flipped classroom: a flipped classroom approach was implemented for both the experimental and control groups;
- Teaching content: the experimental and control groups were taught the same content;
- Grouping: students in the experimental and control groups were randomly assigned to group activities;
- Instructor: the experimental and control groups were taught by the same instructor.

(2) Independent variables:
The study participants were freshmen in two classes of the Information Management Department at Southern Taiwan University of Science and Technology. The experimental and control groups comprised 52 and 47 students in classes A and B, respectively. In each class, students were randomly divided into groups of three or four. The experimental group was offered STUSTCoin rewards as an incentive and were provided with handouts as guidance for group assignments.

(3) Dependant variables:
Three aspects were involved: a comparison of between-group differences in students' intention to preview learning materials; the effect of incentivisation on their class participation and the effect of previewing learning materials; and higher levels of class participation on learning effectiveness.

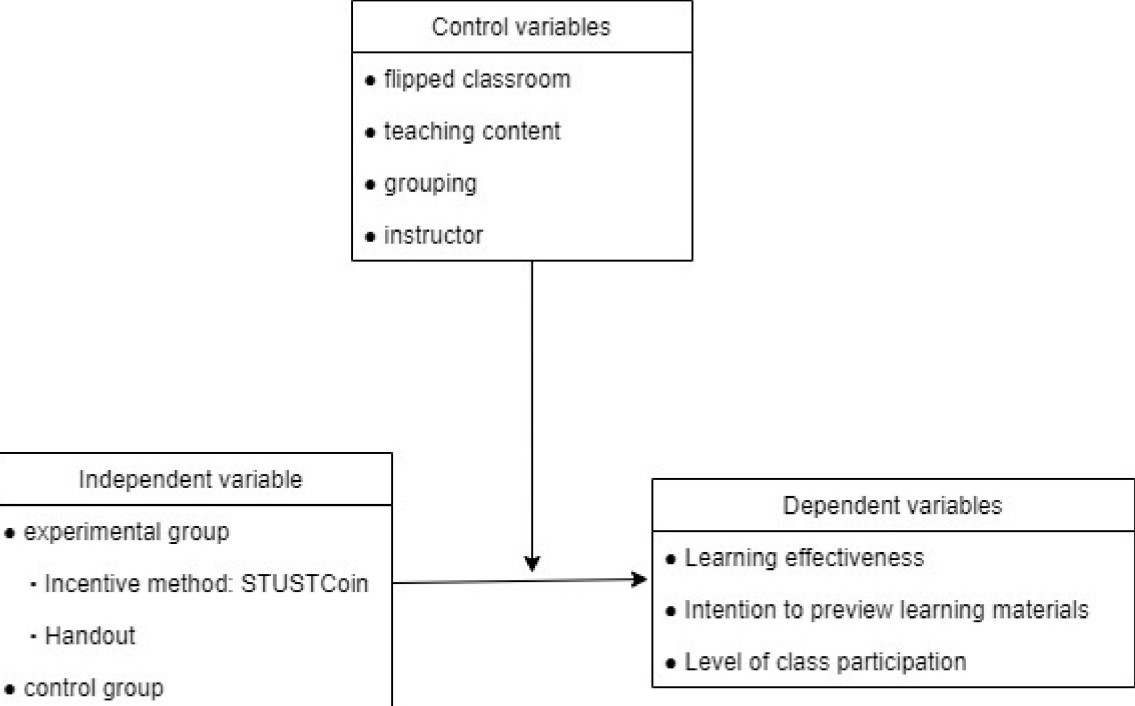

**Figure 1.** Study flowchart.

*2.2. Experiment Design*

2.2.1. Experiment Duration and Procedure

1.  Pre-test questionnaire: experimental groups completed a questionnaire on the intention to preview learning materials;
2.  Experiment procedures: the students attended three flipped classroom-based lessons totalling 150 min every week for 8 weeks. The lessons provided were as follows.
    Step 1. Control group: the students attended 70 min lectures.
    Experimental group:

    *   Each lesson began with a 10 min question-and-answer session. STUSTCoin units were given as rewards, regardless of whether responses were correct;
    *   In the remaining 60 min, the instructor explained the key concepts of the lesson content and randomly asked the students questions, awarding STUSTCoin units to those who participated.

    Step 2. A 20 min class exercise.
    Step 3. The students collaborated by completing 60 min group assignments due by the end of class; these assignments were uploaded to an online learning platform immediately after class. The experimental group was provided with handouts as guidance for the assignments.
3.  At the end of the final lesson, the experimental group completed a questionnaire on the learning platform.

2.2.2. Learning Achievement

An independent-samples *t*-test was used to assess the learning achievements of learners.

2.2.3. Evaluation of Intention to Preview Learning Materials, Classroom Participation, Incentive Method

A questionnaire addressing the three dimensions of the intention to preview learning materials, class participation, and incentivisation with STUSTCoin units was administered, and the mean values were calculated from the valid samples. The study propositions were as follows:

*   P1. A significant difference would be observed in the experimental group's intention to preview learning materials after the intervention;
*   P2. The use of STUSTCoins as incentivisation would lead to significant differences in class participation between the experimental and control groups;
*   P3. Significant between-group differences in learning effectiveness would be noted after the intervention.

2.2.4. Research Tools

The learning platform used was flipClass (flipclass.stust.edu.tw), and all analyses were performed using IBM SPSS Statistics 25. Details are provided as follows:

(1)  Computer classroom
     Because the C# course was conducted in a computer classroom, class activities were performed on computers.
(2)  flipClass learning platform
     The students studied the course material and completed the relevant tasks on flipClass using functions such as the discussion forum and notes. Moreover, the platform included sections on course materials, assignments, and tests.
(3)  Statistical analysis
     IBM SPSS software was used to analyse the data.

## 3. Educational Activity Design

Regarding educational activities, the students were informed in advance of the key topics and concepts for the week. The primary aims of the intervention were to increase

students' intention to preview learning materials and their class participation. The instructor explained the key concepts after randomly asking the students relevant questions. The students then completed group assignments and uploaded them, and these assignments facilitated the comprehension of the material.

*3.1. Lesson Implementation Process*

Figure 2 presents the flowchart of the lesson implementation process.

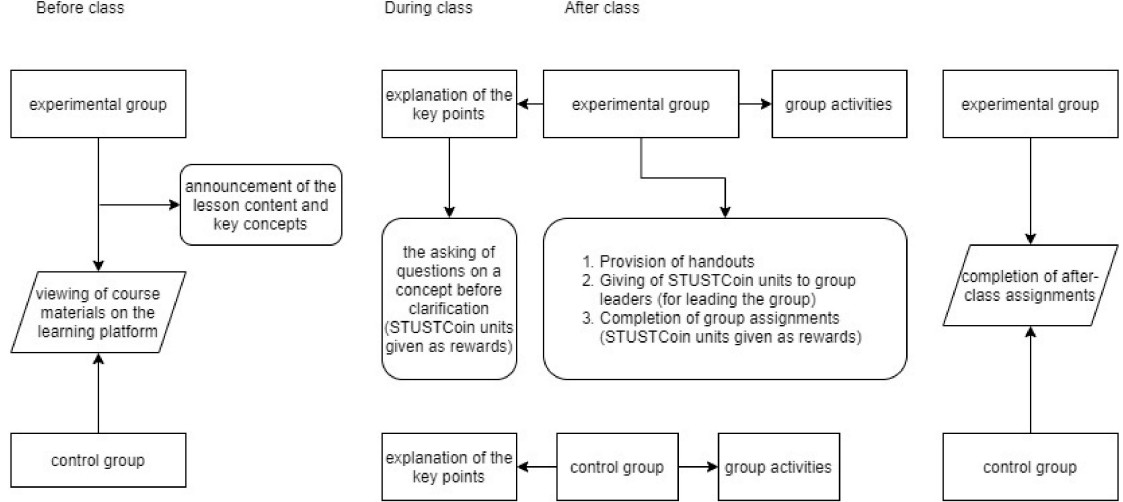

**Figure 2.** Lesson implementation process.

The learning materials were uploaded to flipClass before the start of each lesson. In addition, both groups completed the same group assignments. As mentioned earlier, group assignments needed to be uploaded to the platform immediately after class. Different from the approach used with the control group, for the experimental group, the following were implemented to improve the students' intention to preview learning materials and their class participation:

(1)  At the start of each week, the lesson content and key concepts were uploaded to flipClass to remind the students to make relevant preparations;
(2)  At the start of each lesson, the instructor asked the students questions regarding the lesson content and key concepts. STUSTCoin units were given as rewards regardless of whether the responses were correct;
(3)  The instructor asked questions at random intervals during class. Again, STUSTCoin units were awarded regardless of whether the responses were correct;
(4)  Each group leader was given STUSTCoin units to encourage them to effectively guide the group members to complete group assignments on time;
(5)  Handouts explaining how to complete group assignments were given to the students in the experimental group.

*3.2. Design of the Lesson Content and Handouts*

The contents of the lessons, which were provided during the second semester of the school year, pertained to the C# programming language. Materials, in the form of audio-visual presentations and slideshows, were provided for each of the five main topics, namely, arrays and strings, file and folder management, accessing and writing files and multimedia files, accessing databases, and accessing online materials.

The handouts consisted of four parts: teaching objectives, lesson content, educational activities, and learning steps. The handout template was as follows:

Handout Template

Lesson Topic

1. Teaching objectives
   Explain the lesson themes and learning objectives of the week as well as the concepts that must be understood to achieve these objectives.
2. Lesson content
   The links to the relevant slideshows, videos, and learning resources were given on flipClass.
3. Educational activities
   (1) Viewing the materials: the students were instructed to note the key concepts they identified and any questions they had when viewing the learning materials.
   (2) Explaining the key concepts:
   - The instructor randomly selected students to answer questions during the explanation of each concept, and the instructor awarded them STUSTCoin units;
   - The students were permitted to ask questions about the lesson content during the explanation.
   (3) Group discussion.
   (4) Randomly assigned groups discussed and completed the assignments.
4. Learning guidance
   The main objective was to systematically guide the students through the learning process, including the completion of the assignments.

   The multimedia handout was as follows:
   Topic: Multimedia Files

1. Teaching objectives: understanding the application of multimedia players.
   (1) Learn to use C# to play videos and sound effects;
   (2) Understand how to open/save music files using MenuStrip;
   (3) Understand how to add/delete music files using MenuStrip;
   (4) Understand how to store, update, and delete data using ListBox.
2. Lesson content
   The links to the relevant slideshows, videos, and learning resources were provided on flipClass.
3. Educational activities
   Figure 3 shows the group assignment regarding multimedia players, which is described as follows:
   (1) Subject: multimedia players.
   (2) Required tools: MenuStrip, ListBox, OpenFileDialog, and SaveFileDialog.
   - The multimedia data can be input from a file to ListBox (upper left corner), and the corresponding multimedia file can be played by selecting one of the data entries. Additions and deletions can be made to the multimedia data in ListBox, which can then be saved back to the file;
   - The multimedia data can be generated in ListBox (bottom right corner), and the file can be played by selecting one of the data entries.
4. Learning guidance
   (1) Design related layouts according to the graph, primarily using the design functions in MenuStrip;
   (2) Use OpenFileDialog to open the music file;
   (3) Use SaveFileDialog to save the music file;
   (4) Use ListBox to add and delete the file.

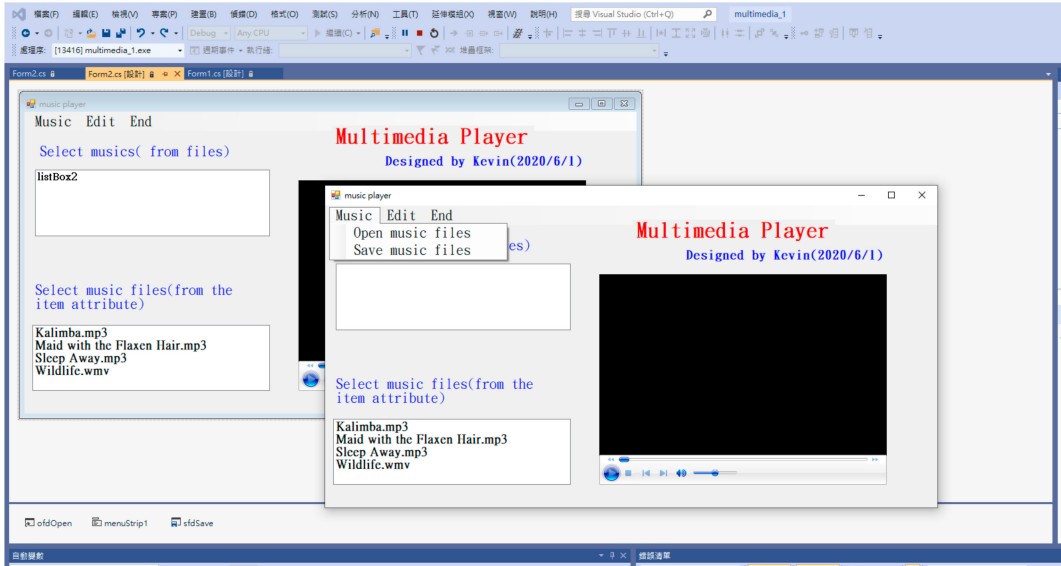

**Figure 3.** Group assignment regarding multimedia players.

## 4. Results

Before the start of the experiment, the students' prerequisite knowledge with regard to the learning material was determined. An independent-samples *t*-test was used to analyse the pre-test results of the two groups; the results are shown in Table 1. The mean pre-test score of the experimental group was 60.87, with a standard deviation (SD) of 13.987; the mean pre-test score of the control group was 59.44, with an SD of 15.660. The *p*-value of 0.624 was nonsignificant, with significance set at a *p*-value of 0.05, indicating that the pre-test scores of the experimental and control groups did not differ significantly; hence, the groups' basic capabilities were the same.

**Table 1.** Pre-test analysis (independent-samples *t*-test analysis).

|  | No. | Mean | SD | *t* |
|---|---|---|---|---|
| Experimental | 52 | 60.87 | 13.987 | 0.492 |
| Control | 47 | 59.44 | 15.660 | |

$p > 0.05$.

Of the 52 questionnaires concerning the students' intention to preview learning materials that were distributed to the experimental group before the intervention, 41 were valid. As shown in Table 2, only 15 students (approximately 36.6%) previewed the class material.

**Table 2.** Pre-intervention for students' intention to preview learning materials (in the experimental group).

| Previewed Learning Materials | No. | Percentage |
|---|---|---|
| YES | 15 | 36.6% |
| NO | 26 | 63.4% |

### 4.1. Analysis of Learning Outcomes

4.1.1. Analysis of the Pre-Test and Post-Test Results of the Control Group

The pre-test and post-test results of the control group were evaluated using a paired-samples *t*-test, and the results are shown in Table 3. The mean pre-test score was 59.64, and the mean post-test score was 71.11. The result of $p = 0.000$ (<0.001) indicated statistical significance and thus implied a considerable difference in the pre-test and post-test scores of the control group.

**Table 3.** Analysis of the paired-samples *t*-test results of the control group.

| Control | No. | Mean | SD | *t* |
|---|---|---|---|---|
| Pre-test | 47 | 59.40 | 15.660 | −18.241 |
| Post-test | 47 | 71.11 | 12.450 | |

*p* < 0.001.

#### 4.1.2. Analysis of the Pre-Test and Post-Test Results of the Experimental Group

The pre-test and post-test results of the experimental group were evaluated using a paired-samples *t*-test, and the results are shown in Table 4. The mean pre-test score was 60.87, and the mean post-test score was 78.73, leading to $p = 0.000$ (< 0.001). This statistically significant result indicated considerable differences in the pre-test and post-test scores of the experimental group.

**Table 4.** Analysis of the paired-samples *t*-test results of the experimental group.

| Experimental | NO. | Mean | SD | *t* |
|---|---|---|---|---|
| Pre-test | 52 | 60.87 | 13.897 | −24.194 |
| Post-test | 52 | 78.73 | 10.202 | |

*p* < 0.001.

#### 4.1.3. Post-Test Analysis

An independent-samples *t*-test was used to evaluate the post-test results of the experimental and control groups to determine the difference between them after the intervention (Table 5). The post-test score of the experimental group was 7.61 points higher than that of the control group, and $p = 0.001$ (<0.01) indicated statistical significance.

**Table 5.** Analysis of the independent-samples *t*-test results of experimental and control groups.

| | No. | Mean | SD | *t* |
|---|---|---|---|---|
| Experimental | 52 | 78.73 | 10.202 | 3.345 |
| Control | 47 | 71.11 | 12.450 | |

*p* < 0.01.

The results validated P3; learning effectiveness differed significantly between the experimental and control groups after the intervention, which involved educational activities and incentivised learning.

### 4.2. Questionnaire Analysis

This study employed a questionnaire for qualitative analysis; 52 questionnaires were distributed, of which 43 were valid, with a valid response rate of 83%.

#### 4.2.1. Reliability Analysis

Table 6 presents the analysis results. All dimensions attained an $\alpha$ value higher than 0.7. The overall scale received an $\alpha$ value of 0.919, implying a certain degree of reliability.

**Table 6.** Reliability analysis of the questionnaire.

| Subscale Name | No. of Items | Cronbach's $\alpha$ |
|---|---|---|
| Intention to preview learning materials | 5 | 0.867 |
| Intention to preview class materials | 5 | 0.910 |
| Incentivisation with STUSTCoin units | 5 | 0.980 |
| Total | 15 | 0.919 |

#### 4.2.2. Analysis of Descriptive Statistics

Table 7 shows the results of the questionnaire for the dimension of the students' intention to preview learning materials. Mean scores of all the items were higher than 3.5,

and the total mean score was 3.911. This result demonstrated that the learners achieved a high intention to preview learning materials.

**Table 7.** Intention to preview class materials.

| No. | Item | Mean | SD |
|---|---|---|---|
| T1. | I want to preview the learning materials because the instructor provides us with STUSTCoin units as rewards. | 3.918 | 0.734 |
| T2. | I preview the learning materials because the instructor provides us with STUSTCoin units as rewards. | 3.898 | 0.821 |
| T3. | I think STUSTCoin rewards motivate me to preview the learning materials. | 3.918 | 0.687 |
| T4. | I preview the learning materials as preparation for next week. | 3.837 | 0.687 |
| T5. | My intention to preview the class materials increases when I am informed of which parts of the materials contain the content and key concepts that will be addressed next class. | 3.814 | 0.732 |
| Overall mean | | 3.911 | 0.747 |

In addition, to compare the students' intention to preview learning materials before and after the intervention, we calculated the percentages of *agree* responses to the questionnaire items, with the responses of *strongly agree* and *agree* considered as *agree* (Table 8).

**Table 8.** Intention to preview class materials (%).

| Item Code | Number of Agreements | Percentage |
|---|---|---|
| T1. | 35 | 71.4% |
| T2. | 35 | 71.4% |
| T3. | 34 | 69.4% |
| T4. | 29 | 67.5% |
| T5. | 27 | 62.8% |
| Mean | 32 | 68.5% |

The total percentage of *agree* responses for items comparing the students' intention to preview learning materials was 68.5%, which was considerably higher than that of 36.6% before the experiment (Table 2). Thus, P1 was validated based on the results in Tables 7 and 8.

Table 9 shows the questionnaire results for the dimension of class participation. The total mean score was 3.65. This result showed that learners achieved high class participation. Therefore, P2 was validated.

**Table 9.** Class participation.

| No. | Item | Mean | SD |
|---|---|---|---|
| C1. | I actively express my opinion during explanations of key concepts. | 3.348 | 0.752 |
| C2. | I ask questions about what I do not understand during explanations of key concepts. | 3.418 | 0.698 |
| C3. | I share my acquired knowledge with my group members during the handout-guided activities. | 3.767 | 0.750 |
| C4. | I propose ideas and suggestions during the handout-guided activities. | 3.790 | 0.773 |
| C5. | I collaborate with my group members to solve problems during the handout-guided activities. | 3.930 | 0.703 |
| Overall mean | | 3.650 | 0.735 |

To further explore whether incentivisation with virtual currency improved class participation, we designed and administered a questionnaire (Table 10). Mean scores for all items exceeded 3.5, and the total mean score was 3.613. This result showed that learners exhibited improved class participation.

**Table 10.** Incentivisation with STUSTCoin units.

| No. | Item | Mean | SD |
|---|---|---|---|
| S1. | My level of class participation (during explanations of key concepts) increases under STUSTCoin incentivisation. | 3.558 | 0.958 |
| S2. | Receiving STUSTCoin units as rewards after in-class quizzes increases my level of class participation. | 3.604 | 0.979 |
| S3. | Receiving STUSTCoin units as rewards for asking questions during the handout-guided group activities increases my participation. | 3.651 | 0.896 |
| S4. | Receiving STUSTCoin units as rewards for sharing knowledge through the handout-guided activities increases my group participation. | 3.651 | 0.973 |
| S5. | Receiving STUSTCoin units as rewards for collaboration with my group members increases my participation in these activities. | 3.604 | 0.954 |
| Overall mean | | 3.613 | 0.952 |

## 5. Discussion and Conclusions

### 5.1. Discussion

Analyses in this study yielded the following findings: regarding the learning outcome, P3 was validated based on the experimental results in Tables 1–4. Regarding the dimension of the intention to preview learning materials, P1 was validated based on the results in Tables 7 and 8; the mean scores of item T1 ("I want to preview the learning materials because the instructor provides us with STUSTCoin units as rewards") and item T3 ("I think STUSTCoin rewards motivate me to preview the learning materials") were 3.918, which were the highest on average, indicating that the students believed that the incorporation of incentivisation into educational activities increased their preview intention. By contrast, the mean score of item T5 ("[m]y intention to preview the class materials increases when I am informed of which parts of the materials contain the content and key concepts that will be addressed next class") was the lowest (3.814). This may be because the students mainly previewed the material so that they would receive STUSTCoin rewards in class. Thus, the students were more motivated to preview the learning materials when incentivised with rewards than simply to inform themselves of the key concepts that would be tested. Regarding the dimension of class participation, P2 was validated based on the results in Table 9; the mean score of item C5 ("I collaborate with my group members to solve problems during the handout-guided activities") was the highest (3.930), demonstrating the overall high participation of the experimental group in group activities as well as the high motivation to submit the assignments on time. By contrast, the mean scores of item C1 ("I actively express my opinion during explanations of key concepts") and item C2 ("I ask questions about what I do not understand during explanations of key concepts") were lower. This is probably because the students had learned a part of the lesson content through preview and were thus less willing to express their opinions or ask questions during explanations of key concepts. Among the five items in the dimension of incentivisation with STUSTCoin units, the mean scores of item S3 ("[r]eceiving STUSTCoin units as rewards for asking questions during group activities increases my participation") and item S4 ("[r]eceiving STUSTCoin units as rewards for sharing knowledge through the handout-guided activities increases my group participation") were the highest. Hence, the incentivisation method increased group participation. The mean score of item S1 ("[m]y level of class participation [during explanations of key concepts] increases under STUSTCoin incentivisation") was the lowest, indicating that the influence of incentivisation during the explanations of key concepts was not high. This is probably because the

questions were answered by higher-level students, giving some students no chance to answer, thereby indirectly affecting their participation.

*5.2. Conclusions*

The authors [15] have reported their evaluation of the impact of a flipped classroom approach on the learning experience of students undertaking an undergraduate biology course. They pointed out that the refinements of components of the flipped design, such as the pre-recorded lectures and the structure of the in-class sessions, may further enhance the student learning experience in this course. Therefore, educational activities and incentivisation were implemented in the flipped classroom context in the present study to increase the students' intention to preview learning materials and their class participation. The results are summarised as follows. The intervention effectively increased this intention, as indicated by the increase from 36.8% to 68.5% in the percentage of the participants in the experimental group who previewed the learning materials after the eight-week intervention. The handouts effectively guided students' learning: class participation was promoted by the handouts combined with incentivisation, particularly in learning activities such as question-and-answer sessions and competition-based tasks. Regarding learning effectiveness, significant between-group differences in the performance of the experimental and control groups (pre-intervention) were observed (i.e., the mean score of the experimental group was significantly higher than that of the control group). This finding indicated that the increases in the students' intention to preview learning materials and their class participation led to corresponding improvements in peer assistance, discussion, and collaboration in problem solving, further enhancing learning effectiveness.

Because the participants were all students from one department of one university, the present results may not necessarily be generalisable to students at other institutions. The experiment was conducted over only one semester; thus, the experimental results must be validated over the long term. Therefore, in the future, follow-up studies should be conducted for the assessment of long-term learning outcomes. In the present study, data collection was slow because of the use of pre-intervention and post-intervention questionnaires. Data collection can be optimised—with regard to both efficiency and data accuracy—if students' data can be automatically recorded on self-built learning platforms. Notably, incentivisation with STUSTCoin units improved both the intention to preview learning materials and class participation. Thus, other forms of intrinsic and extrinsic incentivisation could be integrated into educational activities to determine whether the intention to preview learning materials and class participation can be further enhanced.

**Author Contributions:** Conceptualization, Y.-H.C.; methodology, Y.-H.C.; software, J.-Y.L. and Y.-T.L.; validation, Y.-H.C., J.-Y.L., and Y.-T.L.; writing—original draft preparation, Y.-H.C. and J.-Y.L.; writing—review and editing, Y.-H.C. All authors have read and agreed to the published version of the manuscript.

**Funding:** This research received no external funding.

**Institutional Review Board Statement:** Not applicable.

**Informed Consent Statement:** Not applicable.

**Data Availability Statement:** Not applicable.

**Conflicts of Interest:** The authors declare no conflict of interest.

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
