# Peer review of "Enhancing the Intention to Preview Learning Materials and Participate in Class in the Flipped Classroom Context through the Use of Handouts and Incentivisation with Virtual Currency"

_sustainability, doi:10.3390/su13063276_

Round 1
Reviewer 1 Report
This article is very compelling territory and has great potential.
I believe it can be shaped into something special.
- This is as much a psychological experiment as it is a pedagogical experiment. The incentives are testing intrinsic vs extrinsic motivation and those aspects are very important to include.
- In some ways, the authors are overstating the case here for the flipped component. a) there are several variables in play, b) the benefits of flipped are potentially outweighed by the incentives provided and c) the case for learning is overstated, when it may be short-term memory as assessed by inauthentic assessments. This doesn't mean the piece does not have potential. It does.
- The title is too long and cumbersome. Shorter is better. Is this about flipped or incentives.
- The English needs a good editing for clarity.
- We need references in paragraph one.
- I suggest more background on flipped classrooms. This is not a new invention. What is different is cloud-based tools that enable teachers to create pre-learning in newer ways (now 20 years old).
- This is a study of pedagogy and I think that is what it should do first. How has the pedagogy changed?
- The psychological issues of incentives needs to be explored.
- There are risky ethical issues here about incentivising learning that might be explored.
- The whole internet banking section is unclear to those who won't know about this. And is this pedagogy actually sound.
- The learning claims need to be followed up upon for authentic assessment of learning outcomes in the long-run.
There are many good parts to this to tease out and get clarified. I am still not sure pedagogically what this contributes to the literature.
Reviewer 2 Report
Suggestions to the article. This article presents an interesting experience, however, its formal aspects can be greatly improved. 1. Summary: to. The summary explains the process synthetically, but does not report what you get. 2. In the introduction: to. There is no clear explanation about the learning method adopted, which authors defend it and why they are chosen. b. There is a lack of foundation in the sources of authority, and those that are cited are used to describe their experiences and not to support the arguments of the work. c. There is no expression of the objectives, which are cited below. d. There is also no definition of the variables in the theoretical introduction. 3. The methodology: It does not follow a logical order, as required by a scientific document. Each element of the methodology is not well expressed, in addition they are mixed with each other, which gives rise to confusion. They look themselves: to. The type of methodology used and its justification are missing. b. It refers to objectives that have not been previously expressed. c. The sample has no population of origin, nor does the sampling method. d. Design is confused with procedure. and. The procedure is not well understood because it is not clearly stated. F. There is no clear allusion to how the materials were decided and the materials used. g. The results are not ordered around the hypotheses to be verified. 4. The conclusions must be consistent with the results. 5. There is no discussion, since the sources of authority that contrast with the conclusions of the work are ignored. 4th.- Conclusions: In accordance with and in order to the hypotheses raised. 5th.- Discussion: There are practically no sources on which the conclusions of the work are based. 6th.- Bibliography: Bibliographic sources are scarce.
Reviewer 3 Report
The paper is well-written and presents a very novel idea in student learning environment. I find few minor issues such as the abstract's later part is too short and should incorporate some selected policy and research insights. The last two sections are too short as well as deviod of some links to the existing literature. The methods should elaboratively present the sample selection as well as the steps to remove any bias in this process. The discussion section should present some more insights as well as any limitations and future research needs.
Round 2
Reviewer 1 Report
The authors deflect the reviewers' comments to future research.
The English is still in need of a better edit.
More importantly, this piece merges two big ideas...flipped classrooms and incdentivising learning through the use of the bitcoin rewards. Merging these two makes the claims more iffy. This piece does have good merit. It needs more work to be publishable.
This is primarily an intrinsic v extrinsic motivation piece in the end. It is also about "flipping."
More work to do on this, but it has good merit.
Each lesson began with a 10-minute question-and-answer session. STUSTCoin 165 Sustainability 2021, x, x FOR PEER REVIEW5of 14units were given as rewards regardless of whether the responses were correct.Author Response
Please see the attachment.

Reviewer 2 Report
work has improved significantly, however there are still aspects that are not adequate. In this case, we refer to the discussion that does not exist and that can be implemented, making a comparison of the results with the scientific literature available in this field. This is a very important question considering that the sample does not allow generalization of the results. On the other hand, although of minor importance there is a repetition of phrases in lines 40-41-42-43-44. I encourage you to improve those aspects.
Reviewer 3 Report
Authors have aptly addressed my and other reviewers' comments
